# Solving HNP with One Bit Leakage: An Asymmetric Lattice Sieving Algorithm

**DOI:** 10.3390/e25010049

**Published:** 2022-12-27

**Authors:** Wenhao Shi, Haodong Jiang, Zhi Ma

**Affiliations:** 1State Key Laboratory of Mathematical Engineering and Advanced Computing, Zhengzhou 450001, China; 2Henan Key Laboratory of Network Cryptography Technology, Zhengzhou 450001, China

**Keywords:** HNP, BKZ reduction, sieving, side-channel attack, ECDSA

## Abstract

The Hidden Number Problem (HNP) was introduced by Boneh and Venkastesan to analyze the bit-security of the Diffie–Hellman key exchange scheme. It is often used to mount a side-channel attack on (EC)DSA. The hardness of HNP is mainly determined by the number of nonce leakage bits and the size of the modulus. With the development of lattice reduction algorithms and lattice sieving, the range of practically vulnerable parameters are extended further. However, 1-bit leakage is still believed to be challenging for lattice attacks. In this paper, we proposed an asymmetric lattice sieving algorithm that can solve HNP with 1-bit leakage. The algorithm is composed of a BKZ pre-processing and a sieving step. The novel part of our lattice sieving algorithm is that the lattice used in these two steps have different dimensions. In particular, in the BKZ step we use more samples to derive a better lattice basis, while we just use truncated lattice basis for the lattice sieving step. To verify our algorithm, we use it to solve HNP with 1-bit leakage and 116-bit modulus.

## 1. Introduction

A lattice is a discrete subgroup of $R^{m}$, and is usually presented by a basis. There are infinitely many basis for a non-trivial lattice and we are usually interested in a basis with a short norm and that is orthogonal to other basis, which we call a good basis. Lattice reduction algorithms are designed to find high quality lattice basis, such as LLL reduction and BKZ reduction. The LLL reduction algorithm can be performed in polynomial time and outputs a LLL-reduced basis, which will be shorter and more orthogonal than the original basis. If you want a better lattice basis, then a stronger lattice reduction should be performed, which is what the BKZ reduction algorithm does. The BKZ algorithm is a generalization of the LLL algorithm with a higher block size that can output a much better lattice basis than LLL, and with costs that are exponential with time. With a good basis, we can solve hard problems in lattice with less effort, such as finding the short(est) vector in a lattice or the closest vector to a given target, called SVP and CVP, which are two hard problems in lattice. SVP asks to find the non-zero shortest vector in a given lattice, while CVP asks to find the closest lattice vector to a given target, in a given lattice. To find the short(est) vector in a lattice, there are currently four main methods we can use: enumeration [1,2,3], sieving [4,5,6,7,8], Voronoi cell [9], and Gaussian sampling [10]. Enumeration costs are exponential (of the dimension) in time but polynomial in memory. The best enumeration costs $2^{0.029d^{2}+o\left(d\right)}$ in time. Sieving costs are exponential in both time and memory but the asymptotic time complexity is $2^{0.292d+o(d)}$ for the best sieve algorithm, which is much lower than enumeration in a high dimension. In short, sieving is faster than enumeration when the dimension is larger than 80, approximately. The closest vector problem can be solved via Kannan’s embedding technique, which changes the closest vector problem into a shortest vector problem with 1 more dimension.

Breaking (EC)DSA and Diffie–Hellman with side-channel attacks usually results in a Hidden Number Problem (HNP), which can be converted into a shortest vector problem and solved by SVP algorithm. The Hidden Number Problem is proposed by D.Boneh and R.Venkatesan in 1996 [11] to analyze the bit-security of the private key in some key exchange schemes, such as the Diffie–Hellman key exchange scheme. Later, P. Q. Nguyen and I. E. Shparlinski analyzed the security of Digital Signature Algorithm (DSA) with partially bit-leakage in the private key by HNP. There are two main methods used to solve HNP: the original approach is due to Bleichenbacher and relies on Fourier analysis technique [12]. Another method is a lattice attack, which was discovered by Boenh and Venkatesan in Ref. [11], in which they convert the HNP into a CVP and use the LLL algorithm together with Babai’s nearest plane algorithm [13] to solve it. The time and memory consumption of Bleichenbacher’s method are higher compared with the lattice attack, since Bleichenbacher’s method needs exponential many samples while it only needs polynomial many samples for a lattice attack; however, the Bleichenbacher’s method can solve HNP with less known bits, such as HNP with only 1-bit leakage. However, as for the lattice attack, it is believed that with only 1-bit leakage, a lattice attack has difficulty succeeding [12,14], which is mainly because the lattice constructed by the adversary increases quickly to an unacceptable dimension with the decrease of the known bits.

In the general case of the Hidden Number Problem, the adversary knows some of the most significant bits of the hidden number multiples and some randomly sampled integers modulo for a given integer, which can be translated into modular equations for the hidden number. The hardness of HNP is mainly determined by the size of the modulus and the number of bits known to the adversary. When a lattice attack is applied to HNP, the number of samples affects the distance between the target vector and the lattice, and the dimension of the lattice is nearly the same as the number of samples used. A BDD solver is believed to succeed when the norm of the target vector is less than the shortest vector in the lattice, i.e., ||v||≤GH. When sieving is applied, the constraint on samples used can be relaxed by a scalar factor $\sqrt{4/3}$, that is $\left||v|\right|\leq \sqrt{4/3}\cdot GH$. Since sieving is “more than SVP”, it outputs all the short vectors with a norm below a bound, thus providing more information. With the development of lattice reduction algorithms, Liu and Nguyen solve 160-bit with 2-bit leakage with BKZ2.0 [15] in 2013. Albrecht and Heninger propose the idea of “predicate” [16] and utilize it with General Sieve Kernel (G6K) [17] to break the records. In 2022, Ref. [18] use bits guessing to solve HNP, for each guess, which is a closest vector problem in the same lattice with a different target. As mentioned above, the less bits are known to the adversary, the harder the HNP instance becomes, since with less bits leakage the adversary needs to construct a lattice with a higher dimension.

**Contributions**. We propose an asymmetric lattice sieving algorithm to solve HNP. We use more samples for BKZ pre-processing step to derive a better lattice basis while using truncated lattice basis for sieving.

Compared with previous lattice sieving methods that do not use a BKZ pre-processing step or just yse the same number of samples for both steps, we use more samples for pre-processing and get a better lattice basis, which can benefit the sieving step. How to improve the lattice attack with more samples is a question, and the first solution to it is introduced by Ref. [18], using more samples to find a special HNP instance, such as an instance with small multipliers. We take advantage of “more samples” by applying them to the pre-processing step after the lattice reduction, as the constraint introduced by each sample will propagate to other rows of the basis, and result in a better lattice basis, which is more orthogonal to each other.

We compare our algorithm with “sieve-pred”, the state-of-the-art algorithm mentioned in Ref. [16]. We estimate the cost of our algorithms in various parameters and list it in Table 1 and Table 2. Our algorithm can solve the problem using less time, and the comparison between our algorithm and the state-of-the art algorithm is listed in Table 3. We also experimentally verified the quality of lattice basis obtained by our BKZ pre-processing step, and compare it with previous methods. It turns out that after our BKZ pre-processing step, the lattice basis is more orthogonal compared with previous methods. We illustrate this result in Section 6. To verify our algorithm, we apply it to HNP with only 1-bit leakage, and successfully solved them with a modulus up to 116-bit. We also successfully solved all the parameters reported in Refs. [16,18], and found that our algorithm needs less time. There are still some parameters that we cannot solve at this moment, mainly because the dimension of the lattice is too large.

## 2. Preliminaries

We use ||·|| to denote the Euclidean norm and $\left|\right|\cdot {\left|\right|}_{\infty }$ for infinity norm. We use $v_{\left[i\right]}$ to denote the *i*th entry of a vector and $A_{i,j}$ for the entry in the *i*th row and *j*th column of the matrix *A*. Index starts from 1 in this work.

### 2.1. Lattices

A lattice Λ in $R^{m}$ is a discrete subgroup. Such a lattice is generated by a basis $B=\left(b_{0},b_{1},\cdots ,b_{d-1}\right)\subset Z^{m}$ of linearly independent integer vectors, as $\Lambda =L\left(B\right)=B\cdot Z^{m}=\left\{B\cdot x:x\in Z\right\}$. We define the volume of a lattice Λ as $Vol\left(\Lambda \right)=\sqrt{det(B\cdot B^{T}}$, where *B* is an arbitrary basis of Λ, volume is a lattice invariant since it is independent of the lattice basis used. We use $\pi_{i}:R^{d}\mapsto span{\left(b_{0},b_{1},\cdots ,b_{i-1}\right)}^{⊥},i=0,1,\cdots ,d-1$ to present the orthogonal projections. Particularly, $\pi_{0}\left(\cdot \right)$ means the identity. We use $B^{*}=\left(b_{0}^{*},b_{1}^{*},\cdots ,b_{d-1}^{*}\right)$ to present the Gram–Schmidt orthogonalization (GSO) of *B*, where the Gram–Schmidt vector $b_{i}^{*}=\pi_{i}\left(b_{i}\right)$. Let $\mu_{i,j}=\frac{\langle \;b_{i},\;b_{i}^{*}\rangle }{\langle \;b_{j}^{*},\;b_{j}^{*}\rangle }$. We use $\lambda_{i}\left(\Lambda \right)$ to denote the *i*th successive minimum, which means the smallest *r* such that Λ has *i* linearly independent vectors of the norm at most *r*. $\lambda_{1}\left(\Lambda \right)$ is the norm of a shortest vector in Λ.

Let $S\subset R^{d}$ be a measurable subset with finite volume, then we can use the Gaussian Heuristic to predicate the number of lattice points in S:(1)$$\left\{S\cap \Lambda \right\}\approx \frac{Vol(S)}{Vol(\Lambda )}$$
when S is a closed hyber ball of dimension *d*, which leads to the predication of the length of a non-zero shortest vector in Λ. We use GH(Λ) to denote the expected length of a non-zero shortest vector in Λ, then GH(Λ) is given by:(2)$$GH\left(\Lambda \right)=\frac{\Gamma {\left(1+d/2\right)}^{1/d}}{\sqrt{\pi }}\cdot {\left(\Lambda \right)}^{1/d}\approx \sqrt{\frac{d}{2\pi e}}\cdot {\left(\Lambda \right)}^{1/d}$$
which is the non-zero shortest vector in a lattice usually estimated by $\lambda_{1}=\sqrt{\frac{d}{2\pi e}}\cdot {\left(\Lambda \right)}^{1/d}$.

### 2.2. Hard Lattice Problems

The Shortest Vector Problem (SVP) and Closest Vector Problem (CVP) are in a center position of lattice problems. Many problems can be transformed into hard problems in lattice, which can thus be solved via lattice algorithms.

**Definition** **1.**
*
**(Shortest Vector Problem (SVP)).**
*
*Given a lattice basis B, we need to find a non-zero shortest vector in Λ(B), i.e., find a vector v∈Λ(B) with $\left||v|\right|=\lambda_{1}\left(\Lambda \right)$*


**Definition** **2.**
*
**(Closest Vector Problem (CVP)).**
*
*Given a lattice basis B and a target vector $t\in R^{d}$, we need to find a lattice vector closest to the target t. There is a reduction from CVP to SVP due to Kannan [1], which we refer to as Kannan’s embedding technique. For a closest vector problem with a lattice basis B and a target vector t, it constructs*

(3)
$$L=\left(\begin{array}{cc} B & 0\\ t & \mu \; \end{array}\right)$$

*where μ is the Kannan’s embedding factor. A recommended value for it is $E(\frac{\left||t-v|\right|}{\sqrt{d}})$. For the vector v, which is closest to t, the corresponding vector (v−t,−μ) is small.*


**Definition** **3.**
*
**(α-Bounded Distance Decoding (${\mathrm{BDD}}_{\alpha }$)).**
*
*Given a lattice basis B and a target vector $t\in R^{d}$ which satisfies $dist\left(L\left(B\right),t\right)<\alpha \cdot \lambda_{1}\left(L\left(B\right)\right)$, it asks to find the lattice vector v∈L(B) which is closest to the target t.*


In this paper, we will transform CVP into SVP by Kannan’s embedding technique, since it is thought to be more efficient.

### 2.3. Lattice Algorithms

**Sieving** [4,5,6,7,8] takes a list of points as input, denoted as L∈Λ, and searches for linear combinations of the points that are short. If the initial list is large enough, then it is believed that SVP can be solved by this process recursively. Each point in the list is sampled in polynomial time in *d*.

Assuming that the distribution of the angles of the lattice points in *L* is the same as the distribution of angles sampled randomly from the unit sphere, Phong Q. Nguyen and Thomas Vidick proposed a heuristic sieving algorithm with time complexity of $2^{0.415d+o(d)}$ and memory complexity of $2^{0.2075d+o(d)}$ [7]. Later, Thijs Larrhoven and Benne de Weger sped it up it with Locality Sensitive Hashing, achieving a time complexity of $2^{0.3366d+o(d)}$ and memory complexity of $2^{0.415d+o(d)}$ [8]. The asymptotically fastest sieve achieves a time complexity of $2^{0.292d+o(d)}$ and a memory complexity of $2^{0.415d+o(d)}$, which is sped up by using the Locality Sensitive Filter [5].

If the linear combination takes *k* points at the same time, it is called *k*-sieve. For example, 2-sieve searches for integer combinations of lattice vectors u,v∈L for u≠±v. In high dimensions, we may use the 3-sieve since it requires less memory compared with 2-sieve, but more time consumption.

**The LLL Algorithm** was developed by A. K. Lenstra, H. W. Lenstra, Jr and L. Lovasz in 1982, which can solve the approximate SVP by achieving an approximation factor of ${\left(\frac{2}{\sqrt{3}}\right)}^{n}$. Given a parameter $\frac{1}{4}<\delta \leq 1$, a lattice basis $B=(b_{0},b_{1},\cdots ,b_{n-1})$ is LLL reduced if the Gram–Schmidt orthogonalization of *B* satisfies $\mu_{i,j}\leq \frac{1}{2}$ for i>j, and $\left(\delta -\mu_{i+1,i}^{2}\right)\cdot \left|\right|b_{i}^{*2}\left|\right|\leq \left|\right|b_{i+1}^{*2}\left|\right|$ (Lovasz conditions). Let α=1/(δ−1/4), then the first vector of a LLL reduced basis satisfies $\left|\right|b_{0}\left|\right|\leq \alpha^{(n-1)/2}\cdot \lambda_{1}\left(\Lambda \right)$. For $\frac{1}{4}<\delta <1$, the LLL algorithm can be computed in polynomial time in the dimension.

**The BKZ Algorithm** was proposed by Schnorr in 1987 [19,20] and can be seen as a generalization variant of the LLL algorithm. It obtains higher quality of the output lattice basis, however, with a running time in exponential in the dimension *d*. The BKZ algorithm uses an oracle that solves SVP in the β dimension “block”, and inserts the short vector to the lattice basis recursively. It first finds the shortest vector in the first block $\pi_{1}\left(b_{1}\right)$ and the shortest in $\pi_{1}\left(b_{1}\right)$ will be inserted to the basis. It then proceeds to the next “block” until it reaches the last “block” $\pi_{d-2}\left(b_{d-1}\right)$, which is called a BKZ-tour. After a BKZ-tour, the algorithm will go to the first block and continue this process until the lattice basis remains unchanged. A small, constant number of BKZ-tour is enough for many applications.

The SVP oracle can be instantiated by enumeration or sieving. When it is instantiated by enumeration, it achieves a running time of $1.02^{\beta^{2}+O\left(\beta \right)}$ and a polynomial memory cost in β. As for sieving, the asymptotic time complexity becomes $2^{0.292\beta +o(\beta )}$ and the memory complexity is $2^{0.2075\beta +o(\beta )}$.

### 2.4. The Hidden Number Problem

In order to study the bit-security of private keys in the Diffie–Hellman key exchange scheme, the Hidden Number Problem (HNP) was first proposed by D. Boneh and R. Venkatesan in 1996 [11], who converted the HNP to CVP, using the LLL algorithm to solve it.

In the Hidden Number Problem, q,l is the fixed number known to the public and α is the secret. Given many random t∈Z, there is an oracle $O_{\alpha }\left(t\right)$ that on inputs *t* outputs a tuple (t,a such that ${\left|\alpha \cdot t-a\right|}_{q}<2^{l}$, where ${\left|x\right|}_{q}$ means the unique number 0≤z<q such that z≡xmodq. Suppose we have queried the oracle *m* times and have *m* tuples $(t_{i},a_{i}),i=1,2,\ldots ,m$, then the problem asks to recover the secret α from these tuples. We will write it as $\alpha \cdot t\equiv a_{i}+k_{i}\;\mathrm{mod}\;q$,$0\leq k_{i}<2^{l}$.

The hardness of HNP is mainly determined by the number of leakage and the modulus size; more precisely, it is determined by $\frac{log_{2}\left(q\right)}{leakage}$. The larger the value is, the harder the HNP is.

An important application of HNP is to mount the side channel attack on (EC)DSA. We will introduce DSA and ECDSA, and then take DSA as an example to explain how to transform it into HNP.

### 2.5. Digital Signature Algorithm (DSA)

In DSA, *p* is the modulus and $g\in Z_{q}^{*}$ is an element of order *q*, with q|p−1. Here is a hash function *H* which maps an arbitrary-length input into $Z_{q}$. The private key is $\alpha \in Z_{q}^{*}$ and the public key is $y\equiv g^{\alpha }\;\mathrm{mod}\;p$.

A DSA signature is composed of two integers *r* and *s*, generated as follows:(4)$$r\equiv (g^{k}\;\mathrm{mod}\;p)\;\mathrm{mod}\;q$$
(5)$$s\equiv k^{-1}\left(H\left(m\right)+\alpha r\right)\;\mathrm{mod}\;q$$
where *k* is a random number in $Z_{q}^{*}$ and is unique for each signature.

In order to verify a signature on given a pair (r,s), one needs to compute
(6)$$g^{H\left(m\right)\cdot s^{-1}}y^{rs^{-1}}\;\mathrm{mod}\;q$$
and check whether it equals to *r*.

### 2.6. The Elliptic Curve Digital Signature Algorithm (ECDSA)

ECDSA is an elliptic curve variant of DSA, and is one of the most used signature schemes. In ECDSA, the private key is a randomly generated large number *x* and the public key is computed by [x]G, where *G* is the base point and the multiplication is the scalar multiplication on an elliptic curve. An ECDSA signature is composed of two integers *r* and *s*, which are computed as follows:(7)risthex−coordinateof[k]G
(8)$$s\equiv k^{-1}\left(H\left(m\right)+xr\right)\;\mathrm{mod}\;p$$
where *p* is the modulus and *k* is a random number that is unique for each signature. We call it nonce, and H(m) is the hash of the message.

### 2.7. (EC)DSA as HNP

In the general case of a side-channel attack against (EC)DSA, some of the most significant bits of the signature nonce *k* will be reveled to the adversary. Without loss of generality, we assume that these bits are zero. We will use DSA as an example to explain how to mount a lattice attack on DSA.

Since $s\equiv k^{-1}\left(H\left(m\right)+\alpha r\right)\;\mathrm{mod}\;q$, we rearrange it and then have αr≡ks−H(m)modq. Write $k=k_{1}+k_{2}$, where $k_{1}$ denotes the known part of the nonce *k*, and without loss of generality we assume that $k_{1}=0$, $k_{2}<2^{l}$ is the unknown part. We then have:(9)$$\alpha r\equiv k_{2}s-H\left(m\right)\;\mathrm{mod}\;q$$
(10)$$\alpha \cdot rs^{-1}\equiv k_{2}-s^{-1}H\left(m\right)\;\mathrm{mod}\;q,k_{2}<2^{l}$$

Let $t=rs^{-1}$ and $a=-s^{-1}H\left(m\right)$, then we have a HNP equation:(11)αt≡a+kmodq

### 2.8. Solving the HNP with Lattices

Recall that we have *m* tuples $(t_{i},a_{i}),i=1,2,\cdots ,m$, satisfying $|\alpha t_{i}-a_{i}{|}_{q}<2^{l}$. Boneh and Venkatesan construct the following lattice basis and solve it via a BDD oracle:(12)$$B=\left(\begin{array}{cccccc} q & 0 & 0 & \cdots & 0 & 0\\ 0 & q & 0 & \cdots & 0 & 0\\ \; & \vdots & \; & \ddots & \vdots \\ t_{1} & t_{2} & t_{3} & \cdots & t_{m} & \frac{1}{q} \end{array}\right)$$
Lattice L(B) is generated by the rows of *B*. The target vector is $t=(a_{1},a_{2},\cdots ,a_{m},0)$ and the lattice vector $v=\left(t_{1}\alpha \;\mathrm{mod}\;q,t_{2}\alpha \;\mathrm{mod}\;q,\cdots ,t_{m}\alpha \;\mathrm{mod}\;q,\frac{\alpha }{q}\right)\in L\left(B\right)$ is close to *t*, with $||t-v||\leq \sqrt{m+1}\cdot 2^{l}$. We will call *v* as the hidden vector since it contains the information about the hidden number α. This method can only solve HNP with large leakage. For small leakages such as 2-bit or 1-bit leakage it will not succeed.

We can solve this BDD problem via CVP methods or use Kannan’ embedding technique to change it into a shortest vector problem.

Martin R.Albrecht and Nadia Heninger use two techniques to improve the attack [16]: the recentering technique and the elimination method. These two techniques play an important role in pushing the boundaries of the unique shortest vector scenario.

The recentering technique is first described in Ref. [21] and provides a significant improvement in practice. It works as follows: since $0\leq k_{i}<2^{l},i=1,2,\cdots ,m$, we can reduce the size of $k_{i}$ by 1 bit via letting $a_{i}^{{}^{\prime }}=a_{i}+2^{l-1},i=1,2,\cdots ,m$, thus $k_{i}^{{}^{\prime }}=k_{i}-2^{l-1},i=1,2,\cdots ,m$. Now we have reduced $k_{i}$ by 1 bit because $-2^{l-1}\leq k_{i}^{{}^{\prime }}<2^{l-1}$.

The elimination method is described in Ref. [16]. It works as follows: since we have *m* equations $a_{i}+k_{i}\equiv \alpha t_{i}m\;\mathrm{mod}\;q,i=1,2,\cdots ,m$, we rearrange these equations and then we have:(13)$$\alpha \equiv t_{1}^{-1}\left(a_{1}+k_{1}\right)\equiv t_{2}^{-1}\left(a_{2}+k_{2}\right)\equiv \cdots \equiv t_{m}^{-1}\left(a_{m}+k_{m}\right)\;\mathrm{mod}\;q$$
for each equation $t_{1}^{-1}\left(a_{1}+k_{1}\right)\equiv t_{i}^{-1}\left(a_{i}+k_{i}\right)\;\mathrm{mod}\;n,i=2,3,\cdots ,m$, we rearrange it to get
(14)$$a_{i}-t_{i}\cdot t_{1}^{-1}\cdot a_{1}+k_{i}\equiv t_{i}\cdot t_{1}^{-1}\cdot k_{1}\;\mathrm{mod}\;q$$
thus we have a new HNP instance with $a_{i}^{{}^{\prime }}=a_{i}-t_{i}\cdot t_{1}^{-1}\cdot a_{1}$ and $t_{i}^{{}^{\prime }}=t_{i}\cdot t_{1}^{-1}$, now the secret is $k_{1}$ and we have m−1 relations about it.

There are two advantages of this method: . it can reduce the dimension of the lattice by 1, also making the secret and the unknown parts $k_{i}$ equal sized.

Let $w=2^{l-1}$, with these two techniques, Martin R. Albrecht and Nadia Heninger construct a new lattice Λ generated by:(15)$$B=\left(\begin{array}{ccccccc} q & 0 & 0 & \cdots & 0 & 0 & 0\\ 0 & q & 0 & \cdots & 0 & 0 & 0\\ \; & \vdots & \; & & \vdots \\ 0 & 0 & 0 & \cdots & q & 0 & 0\\ t_{2}^{{}^{\prime }} & t_{3}^{{}^{\prime }} & t_{4}^{{}^{\prime }} & \cdots & t_{m}^{{}^{\prime }} & 1 & 0\\ a_{2}^{{}^{\prime }} & a_{3}^{{}^{\prime }} & a_{4}^{{}^{\prime }} & \cdots & a_{m}^{{}^{\prime }} & 0 & w \end{array}\right)$$
There is a short vector $v=(w-k_{2},w-k_{3},\cdots ,w-k_{1},w)$ in Λ with norm $\left||v|\right|\leq \sqrt{m+1}\cdot w$. The parameter *w* is actually the Kannan’s embedding factor and a recommended value for it is $E(\frac{\left||t-v|\right|}{\sqrt{d}})$. Furthermore, *w* is also the upper bound of the known bits after using recentering technique, since $-2^{l-1}\leq w<2^{l-1}$.

## 3. Algorithm

We propose a two-step algorithm to solve the HNP. The algorithm is composed of a pre-processing algorithm and a sieving algorithm. The pre-processing algorithm takes *m* samples as input and outputs a BKZ-β reduced basis with $m^{\prime }+1$ dimension, which is smaller than the original dimension m+1.

Compared to only using BKZ reduction, we use sieving to reduce the dimension of the lattice, which is because the success condition of BKZ is different from it than sieving, mainly due to the fact that sieving can produce exponentially many short vectors while BKZ reduction cannot. The difference of dimension between BKZ and sieving is listed in Table 1, and it can be seen that for the parameters considered in this paper, the difference is large, for example, 36 for 1-bit leakage and the 116-bit modulus.

Compared with only using sieving, we add a pre-processing step; the cost is negligible when compared with the sieving step, but it produces a better lattice basis. We experimentally verified the effect of the pre-processing step and find that the basis obtained by our BKZ pre-processing step is more orthogonal than the other BKZ algorithm.

The sieving algorithm will output a list of all the short vectors with a norm smaller than $\sqrt{4/3}GH$ and we will check the list for the desired hidden number α.

### 3.1. Baseline

Assume that we have *m* tuples of $\left(t_{i},a_{i}\right)$. We will use the recentering technique and elimination technique mentioned above to pre-process the HNP instances, and construct a lattice in the same way as Ref. [16]. We then choose a submatrix for it and apply lattice sieving.

### 3.2. Pre-Processing Algorithm

We use more samples to construct lattice basis *B* because it can take advantage of more information about the secret and results in a better basis for solving HNP. In this way, after the BKZ-β reduction, more information about the secret will propagate to every row of the basis, and more constraints are used to the lattice basis.

After the BKZ-β reduction we will choose a $\left(m^{\prime }+1\right)\times \left(m^{\prime }+1\right)$ submatrix of *B*. We choose the last two columns because they contain the information about the secret, since the expected hidden vector is $v=(w-k_{2},w-k_{3},\cdots ,w-k_{m},w-k_{1},w)$. We choose $m^{\prime }-1$ columns in the rest randomly, so the result in the hidden vector comes to $v\;=\;(w-k_{i_{1}},w-k_{i_{2}},\cdots ,w-k_{i_{m^{\prime }-1}},w-k_{1},w)$.

In this way, we have a matrix $B^{\prime }$ that has *m* rows and $m^{\prime }+1$ columns, $m>m^{\prime }+1$. It is clear that $B^{\prime }$ has linear dependence in rows. We will therefore apply the LLL algorithm to it. There are two benefits we can get from the LLL algorithm: first, it can eliminate the linear dependence in rows conveniently, and furthermore, by using it we can get a more orthogonal basis (see Algorithm 1).
**Algorithm 1** Pre-processing algorithm.**Input:** *m* tuples of $\left(t_{i},a_{i}\right)$, parameter $m^{\prime }$ and block size β**Output:** A BKZ −β reduced basis $C_{\left(m^{\prime }+1\right)\times \left(m^{\prime }+1\right)}$1:Construct a lattice basis with $(t_{i},a_{i}),i=1,2,\cdots ,m$, denoted as *B*2:Perform BKZ −β reduction on *B*, which results in a BKZ −β reduced basis $B_{\left(m+1)\times (m+1\right)}^{{}^{\prime }}$3:Sample $m^{\prime }-1$ columns randomly from the first m−2 columns of *B*, create a matrix $C_{m\times (m^{\prime }+1)}$ with the sampled $m^{\prime }-1$ columns together with the last 2 columns of $B^{\prime }$, and all the rows of $B^{\prime }$4:Perform LLL algorithm on $C_{m\times (m^{\prime }+1)}$ to eliminate linear dependence in rows5:Delete the first $m^{\prime }+1-m$ rows, which are all-zero, to obtain a $\left(m^{\prime }+1\right)\times \left(m^{\prime }+1\right)$ matrix *C*6:**return** C

### 3.3. Sieving

We apply the lattice sieving algorithm to the $d=m^{\prime }+1$ dimension lattice L(C). The sieving algorithm will output all the short vectors with a norm less than $\sqrt{4/3}GH$ in L(C), and we check the list for candidates.

We will explain how to choose $m^{\prime }$ to ensure that this algorithm succeeds with a high probability in the next section. We point out that since the hidden vector in L(C) is $v=\pm (w-k_{i_{1}},w-k_{i_{2}},\cdots ,w-k_{i_{m^{\prime }-1}},w-k_{1},w)$, we can therefore recover $k_{1}$ from the $m^{\prime }$th column of *v*, and thus calculate α as described in Algorithm 2, lines 4–5.
**Algorithm 2**  Sieving for HNP.**Input:** A lattice basis *C* of $m^{\prime }+1$ dimension**Output:** The hidden number α for the HNP1:Perform sieving on the lattice L(C) to get a list *L* of all short vectors with norm less than $\sqrt{4/3}GH$2:**for all***v* in the list *L*
**do**3: **if**
$abs(||v|{|}_{\infty })\leq w$ and $abs(v_{\left[m^{\prime }+1\right]})=w$ **then**4:  Compute $k_{1}=\left(w-v_{\left[m^{\prime }\right]}\right)\cdot \left(v_{\left[m^{\prime }+1\right]}/w\right)$5:  Compute $\alpha \equiv t_{1}^{-1}\cdot \left(a_{1}+k_{1}\right)\;\mathrm{mod}\;q$6:  **if** α satisfies all the tuples $(a_{i},t_{i}),i=1,2,\cdots ,m$ **then**7:   **break;**8:   **return(α)**9:  **else**10:   continue;11:  **end if**12: **end if**13:**end for**14:**return**(“Failed.”)

## 4. Analysis

### 4.1. Time Complexity and Memory Complexity

We construct the lattice basis *B* in polynomial time and the BKZ-β reduction can be computed in a running time of $2^{0.292\beta +o(\beta )}$. Random sampling as well as the LLL algorithm will complete in a polynomial time. We will later perform lattice sieving; the cost is $2^{0.292d+o(d)}$ in time for the asymptotically fastest sieving or 0.658·d−21.11·log(d)+119.91 for log of cost in the CPU cycles recommended in Ref. [16].

The result in time complexity of the algorithm is
(16)$$T=2^{0.292\beta +o(\beta )}+2^{0.292d+o(d)}$$
A recommended value of β is β=d−20.

For memory complexity, we should hold a list of vectors output by sieving in the last step and the database for sieving during lattice sieving. The number of vectors output by sieving can be estimated by Gaussian heuristic, namely ${\sqrt{4/3}}^{d}$, and the size of the database for sieving is $O({\sqrt{4/3}}^{d})$. Thus, the asymptotic memory complexity is
(17)$$M=2^{0.2075d+o(d)}$$

### 4.2. Number of Samples

Now we analyze the number of samples needed in the problem. Let us analyze the number of samples for sieving, namely $m^{\prime }$, and the corresponding sieving dimension is $m^{\prime }+1$. It is well known that sieving can output a list of all short vectors with a norm less than $\sqrt{4/3}GH$, so we expect that the hidden vector *v* should be contained in the list if $\left||v|\right|\leq \sqrt{4/3}GH$. If we use the BKZ algorithm to solve the problem, since it can only provide *d* short vectors, the corresponding condition will become ||v||≤GH. This difference will result in a large gap in the dimension of the lattice, so the BKZ algorithm should construct a much larger lattice than sieve. We list the gap in Table 1.

GH can be computed as follows: for the lattice generated by *C*, it is exactly the same lattice generated by randomly choosing $m^{\prime }$ samples and constructing a lattice basis in the same way as with Equation (Equation 15). So, the volume of L(C) is $Vol\left(L\left(C\right)\right)=q^{m^{\prime }-1}\cdot w$, thus the GH is
(18)$$GH\left(L\left(C\right)\right)=\frac{\Gamma {\left(1+\left(m^{\prime }+1\right)2\right)}^{1/(m^{\prime }+1)}}{\sqrt{\pi }}\cdot Vol{\left(L\left(C\right)\right)}^{1/(m^{\prime }+1)}=\sqrt{\frac{m^{\prime }+1}{2\pi e}}\cdot {\left(q^{m^{\prime }-1}\cdot w\right)}^{1/(m^{\prime }+1)}$$

As for the norm of the hidden vector *v*, it can be bounded by $\sqrt{m^{\prime }+1}\cdot w$ since each entry of *v* is bounded by *w*. However, it can be estimated more precisely since it is assumed that $t_{i}$ is distributed uniformly in $Z_{q}$, and α is a random number in $Z_{q}^{*}$, so we can assume that $k_{i}$ is distributed uniformly and randomly in $Z_{q}$. Thus. we can compute the expectation of ||v||, which is the same as [16].
(19)$$E\left(||v||\right)=\sqrt{E(\sum_{i=1}^{m^{\prime }}{\left(w-k_{i}\right)}^{2}+w^{2}}=\sqrt{m^{\prime }\cdot {\left(w-k_{i}\right)}^{2}+w^{2}}=\sqrt{\frac{m^{\prime }\cdot w^{2}}{3}+\frac{m^{\prime }}{6}+w^{2}}$$

Thus, we use the constraint $E\left(v\right)\leq \sqrt{4/3}GH$ to find $m^{\prime }$. That is, we use the minimum $m^{\prime }$ such that
(20)$$\sqrt{\frac{m^{\prime }\cdot w^{2}}{3}+\frac{m^{\prime }}{6}+w^{2}}⩽\sqrt{4/3}\cdot \sqrt{\frac{m^{\prime }+1}{2\pi e}}\cdot {\left(q^{m^{\prime }-1}\cdot w\right)}^{1/(m^{\prime }+1)}$$
holds.

## 5. Experiment on HNP

We apply the two-step algorithm on several HNP instances with only 1-bit leakage. All the experiments are performed by SageMath and G6K [17] on Intel Xeon Platinum 8280 @ 224x 4GHz. We use “uSVP” to present the method of solving HNP via uSVP, such as using the BKZ algorithm to solve HNP, and the corresponding number of samples needed is estimated by E(||v||)≤GH. When sieving is applied, we use $E\left(||v||\right)\leq \sqrt{4/3}\cdot GH$ to estimate the number of samples needed, since “sieving is more than SVP”. Δm stands for the difference of samples needed for uSVP and sieving. It can be seen that sieving can use a much smaller lattice.

We list the number of samples needed for 1-bit leakage. “uSVP” stands for solving HNP by BKZ algorithm and “Sieving” stands for using Algorithm 2. We have solved 1-bit leakage with modulus up to 116-bit, and list the expected requirements for a larger modulus. We point out that the main constraint for larger parameters is the memory consumption, for example, when solving HNP with 1-bit leakage and a 116-bit modulus, the peak memory reached is 960 GB, which is unacceptable for larger parameters.

We solved all the instances listed in Table 2, except for 2-bit leakage with a 256-bit modulus. We need to construct a lattice of dimension 132. The memory cost becomes to the main obstacle with the dimension going up.

Let us take 116-1 HNP as an example. We show how the number of samples $m^{\prime }$ affects E(||v||) and $\sqrt{4/3}\cdot GH$ in Figure 1. The x-axis stands for the number of samples $m^{\prime }$ used for sieving. The y-axis stands for the value of E(||v||) and $\sqrt{4/3}\cdot GH$, since they are functions of $m^{\prime }$. The red line is E(||v||) and the black line is $\sqrt{4/3}\cdot GH$. The crossing point is the value of $m^{\prime }$ we choose to solve HNP. When $E\left(||v||\right)\leq \sqrt{4/3}\cdot GH$, which means that the red line it lower than the black line, the HNP is believed to be solvable, and the corresponding minimum $m^{\prime }$ is the samples we use for sieving.

We take 116-bit modulus with 1-bit leakage as an example and illustrate it in Figure 2. This figure shows how the number of samples affect the gap between $\lambda_{1}\left(\Lambda \right)$ and the expectation of the hidden vector. With an increasing number of samples, the value of $E\left(||v||\right)/\lambda_{1}\left(\Lambda \right)$ decreases, and it becomes solvable when $E\left(||v||\right)/\lambda_{1}\left(\Lambda \right)\leq \sqrt{4/3}$

Regarding the number of samples required for Algorithm 2, the point of intersection is the number of samples that we use. However, we find that the limitation of $E\left(||v||\right)\leq \sqrt{4/3}\cdot GH$ is not a necessary condition. For example, 2-bit leakage with a modulus of 160-bit is expected to be solvable with more samples than 84 but can be solved with only 77 samples with a success probability of nearly 1.

## 6. Comparison of BDD with Predicate

In this section, we compare our asymmetric lattice sieving algorithm with previous lattice methods. To the best of our knowledge, there are two algorithms that achieve the same result: the BDD with the predicate method [16] and the bit guessing method [18]. Experimentally, the BDD with the predicate method is faster and gives a thorough analysis of its algorithm in various parameters. So, we compare our algorithm with the algorithm mentioned in Ref. [16], which is the state-of-the-art algorithm for solving HNP. There are four algorithms mentioned in Ref. [16] for the different parameters: “BKZ-enum”, “BKZ-sieve”, “enum-pred”, and “sieve-pred”. For the parameters considered in this paper, we mainly use the “sieve-pred” algorithm to solve the problems, since “sieve-pred” is the fastest algorithm for these parameters. Therefore, we compare our algorithm with the “sieve-pred” algorithm in Ref. [16].

In this table, “Ours” stands for our asymmetric two-step sieving and “sieve-pred” stands for the “sieve with predicate” algorithm mentioned in Ref. [16]. The time is walltime and all these experiments are performed on the same machine. In the same dimension, our algorithm obtains a better basis in the aspect of orthogonality, since we use more samples to restrict the reduction process. Let us take 2-bit leakage with a modulus of 224-bit as an example. The following figure shows that the basis obtained by our algorithm is more orthogonal. Note that the range of y-axis in “Ours” and “Previous methods” is different.

We experimentally verified the orthogonality of the lattice basis obtained by our BKZ pre-processing step, and find that it is more orthogonal to each other and thus we obtain a better lattice basis.

We demonstrate the conclusion by computing the cosine value between each basis. That is, we first generate two lattice basis: one is obtained by our BKZ-pre-processing step, denoted as $B_{ours}=\left(b_{ours,0},\cdots ,b_{ours,d-1}\right)$, and the other is obtained by the previous method, denoted as $B_{previous}=\left(b_{previous,0},\cdots ,b_{previous,d-1}\right)$. We then calculate the cosine values and compare them. The cosine values are calculated as follows:(21)$$cos_{ours_{i,j}}=\frac{\langle b_{ours,i},b_{ours,j}\rangle }{∥b_{ours,i}∥\cdot ∥b_{ours,j}∥},i,j=0,1,\cdots ,d-1$$
(22)$$cos_{previous_{i,j}}=\frac{\langle b_{previous,i},b_{previous,j}\rangle }{∥b_{previous,i}∥\cdot ∥b_{previous,j}∥},i,j=0,1,\cdots ,d-1$$

We can draw the results based on Figure 3. The x-axis stands for the cosine values and the y-axis stands for the number of the cosine values of the basis. So, these figures show the distribution of the cosine of lattice basis, It can be seen that the basis obtained by our algorithm is more orthogonal since its cosine value is more centered at zero, which means the angle is closer to $\frac{\pi }{2}$. We combine the figure “Ours” and “Previous methods” together to get the figure “Comparison”. In “Comparison”, the red curve stands for the cosine distribution of lattice basis obtained by our algorithm and the blue curve stands for previous methods. It can be seen that the cosine distribution is more centered at zero, which means that the basis is more orthogonal to each other.

However, there are still two problems unsolved: how to choose the pre-processing step parameter *m* and how the angle between the lattice basis affects the sieving step. As for the first question, we usually choose m=2d for simplicity. That is because for the parameters considered in this paper, performing a BKZ-β reduction on a lattice of dimension 2d is acceptable. If we use a large *m*, the pre-processing step will be too expensive. As for the second question, a better basis can make it easier to find “good combinations”, which will give a shorter vector in lattice. However, how the angle distribution affects the sieving step needs more rigorous analysis.

## 7. Conclusions

In this paper, we proposed an asymmetric lattice algorithm for HNP. We call it “asymmetric” since the algorithm uses a different number of samples for the two steps.

Compared with the BKZ algorithms, we use sieving to solve HNP since sieving can reduce the dimension of the lattice significantly, as can be seen in Table 1, parameter Δm. The main reason why sieving can reduce the dimension is that sieving can produce exponentially many short vectors while BKZ algorithms cannot. For the parameters considered in this paper, this reduction in dimension is usually over 20, which results in a significant speedup in time. Compared with sieving only, we apply a BKZ pre-processing step with more samples to make use of the information that each sample sufficiently gives. We thus expect to obtain a better lattice basis, namely, a basis that is more orthogonal.

We experimentally verified the efficiency of our algorithm, and applied it to solve HNP with 1-bit leakage with a modulus up to 116-bit. To verify the effect of the pre-processing step, we studied the “cosine distribution” of the lattice basis obtained by our algorithm and other methods, and conclude that the angle of our basis tends more to $\frac{\pi }{2}$, which means that it is more orthogonal. To verify the overall efficiency of our algorithm, we compared it with the state-of-the-art algorithm mentioned in Ref. [16]. We performed the algorithms on the same machine and compared the runtime. The results can be seen in Table 1 and Table 2.

An analysis of the parameters used in this algorithm is also given. We take 1-bit leakage and a 116-bit modulus as an example, and we illustrate the effect of $m^{\prime }$ in Figure 1 and Figure 2. For the other parameters, we list it in Appendix A. However, for the parameter *m* used for the pre-processing step, we simply choose m=2d. This is because performing a BKZ reduction in a lattice of 2d dimension is acceptable and has a good result on the basis. For smaller *m*, the effect of pre-processing step will be reduced. More rigorous analysis and experimental verification will be done in future work.

## Figures and Tables

**Figure 1 entropy-25-00049-f001:**
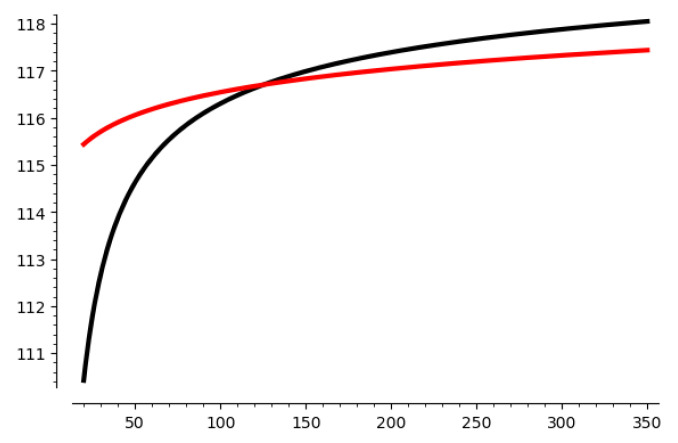
The curve of E(||v||) and $\sqrt{4/3}\cdot GH$ with respect to the number of samples $m^{\prime }$.

**Figure 2 entropy-25-00049-f002:**
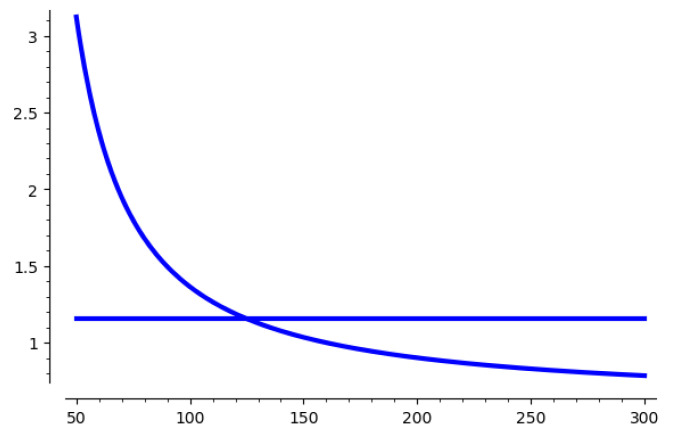
$E\left(||v||\right)/\lambda_{1}\left(\Lambda \right)$ of 116-bit modulus with 1-bit leakage.

**Figure 3 entropy-25-00049-f003:**
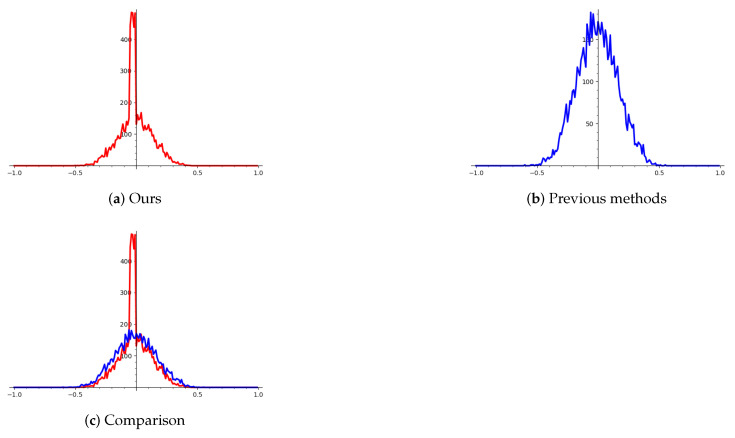
Comparison of the cosine distribution. The x-axis stands for the cosine value and the y-axis stands for the number of angles with cosine equal to x-axis. “Ours” is our pre-processing step and “Previous methods” is the other BKZ methods.

**Table 1 entropy-25-00049-t001:** Resources required to solve HNP with 1-bit leakage.

	80-bit	90-bit	100-bit	112-bit	116-bit	128-bit	160-bit
Samples for uSVP	112	126	139	155	161	177	220
Samples for sieving	87	98	108	121	125	137	171
Δm	25	28	31	34	36	40	49
Sieving dimension	88	99	109	122	126	138	172
Sieving cost	24.6 s	179.1 s	1131.9 s	15,847.5 s	46,598.8 s	-	-

**Table 2 entropy-25-00049-t002:** Resources required to solve HNP.

160-bit		
Leakage	2-bit	3-bit
Sieving dimension	85	57
Sieving cost	47.3 s	<1 s
192-bit		
Leakage	2-bit	3-bit
Sieving dimension	100	67
Sieving cost	606.9 s	3.2 s
224-bit		
Leakage	2-bit	3-bit
Sieving dimension	117	77
Sieving cost	14,850.8 s	8.2 s
256-bit		
Leakage	2-bit	3-bit
Sieving dimension	134	88
Sieving cost	-	59.8 s

**Table 3 entropy-25-00049-t003:** Comparison with “sieve-pred”.We compare the algorithms for HNP with 1-bit and 2-bit leakage with various modulus. The “Time” stands for the average time.

		1-bit Leakage				2-bit Leakage		
	**log${}_{2}$(q)**	**80-bit**	**90-bit**	**100-bit**	**112-bit**	**160-bit**	**192-bit**	**224-bit**
Ours	Time	24.6 s	179.1 s	1131.9 s	15,847.5 s	47.3 s	606.9 s	14,850.8 s
	Samples	87	98	108	121	84	99	116
sieve-pred	Time	25.1 s	228.7 s	3868.4 s	18,206.8 s	51.7 s	743.2 s	29,616.1 s
	Samples	87	98	108	121	87	98	116

## Data Availability

The data presented in this study are available within the article.

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
