# Peer review of "Solving HNP with One Bit Leakage: An Asymmetric Lattice Sieving Algorithm"

_entropy, 2022, doi:10.3390/e25010049_

Round 1
Reviewer 1 Report
The manuscript can be accepted post appropriate revisions are made:
1. The authors should remove the word "novel" from the manuscript. Whether anything is novel or not is for the scientific community to decide and not the authors.
2. The introduction is too short. The authors need to put the problem into a bit more context. The amount of previous work done as written by the authors shows that "either no one was interested to solve this problem" or "this is not a problem to solve at all". A bit more literature search is needed.
3. "We use || · || to denote the Euclidean norm and || · || for infinity norm". I am a bit confused here, do they use the same notation for euclidean and infinity norm? If yes, please change it for better understanding.
4. The authors are advised to put a separate conclusion section. The last paragraph on future work needs to go there.
5. No need to put Algorithm 3 in the paper since that is already a published paper. The author can remove it and just place a reference. Or create an appendix or put it in a supplementary document.
6. Is there any limitation to the proposed algorithm? Please provide something on this in the manuscript.
7. The authors cannot make statements like "is still unclear". They should still provide some insight that they have and say that its verification and rigorous understanding will be part of future work.
Reviewer 2 Report
In the manuscript, the authors propose a composite lattice sieving algorithm to solve the Hidden Number Problem (HNP), which is usually used to hack the Diffie-Hellman key exchange scheme. They combine the idea of the former BKZ algorithm and sieving-based technique. As a result, they find the new algorithm can find a better lattice-basis solution in HNP. Using this new algorithm, they test its ability to solve HNP with 1-bit leakage.
While the new algorithm and results are interesting, I still have some concerns on the significance and contribution of the manuscript,
1. What is the major advantage of this BKZ-sieving algorithm comparing to the original BKZ algorithm in [3,10,11]? It seems that the time-complexity and memory-complexity of the BKZ algorithm (in line 119) and the new algorithm (in Eq. (16) and (17)) are the same.
2. Here, the authors mention that they use more samples to solve the lattice problem compared with former experiments. Will the improvement of finding the lattice basis originate from the increasing of the sample number?
3. One major claim here is that the new algorithm can solve HNP with 1-bit leakage while the former algorithms are believed not able to. However, all the algorithms (including the new algorithm here) suffer from an exponentially increasing time and memory complexity. From this point, I cannot see why the authors claim that their new algorithm can solve HNP with 1-bit leakage while former algorithms cannot. Maybe a better claim in the abstract is that “the new algorithm can solve HNP with 1-bit leakage with modulus up to 116 bits”.
If the authors address the following problems properly, I can suggest its publication. Below I also have some minor comments,
1. The authors should define the acronyms for SVP, CVP, LLL, BKZ at their first occurrence in the introduction.
2. In line 58, there is a typo, “successf”.
3. In line 228 and line 235, the authors should present the numerically results with a better manner. What are the black and red lines? Where is the x and y axis labels? The authors should also add captions and apply larger font sizes for the x and y axis labels.

Reviewer 3 Report
Minor issues:
-
There are figures on pages 9, 10, and 11 without numbers and captions. All the figures in the manuscript have to be numbered.
-
Please enlarge Figure 1 on page 9. It is too small.
-
Captions for tables must be on top, not bottom.
-
References are being cited out of order. All the citations need to be in ascending order.
-
On page 2: “We use || · || to denote the Euclidean norm and || · || for infinity norm.”
Is there any mistake? Because the two denotations are the same.
Major issues:
-
I found that the paper still lacks comparison. The only comparison was with the [1]. Could you include more recent works in the comparison tables?
Round 2
Reviewer 1 Report
The authors made satisfactory revisions to the manuscript. I am happy to recommend the publication of the manuscript in its current form.
Reviewer 3 Report
The authors have fixed all the previous concerns. This version of the manuscript is ready to publish.